# Implications of being born late in the active season for growth, fattening, torpor use, winter survival and fecundity

Britta Mahlert[1], Hanno Gerritsmann[1], Gabrielle Stalder[1], Thomas Ruf[1], Alexandre Zahariev[2,3], Stéphane Blanc[1,2,3], Sylvain Giroud[1]*

[1]Department of Integrative Biology and Evolution, Research Institute of Wildlife Ecology, University of Veterinary Medicine, Vienna, Austria; [2]Université de Strasbourg, IPHC, Strasbourg, France; [3]CNRS, UMR7178, Strasbourg, France

**Abstract** For hibernators, being born late in the active season may have important effects on growth and fattening, hence on winter survival and reproduction. This study investigated differences in growth, fattening, energetic responses, winter survival and fecundity between early-born ('EB') and late-born ('LB') juvenile garden dormice (*Eliomys quercinus*). LB juveniles grew and gained mass twice as fast as EB individuals. Torpor use was low during intensive growth, that are, first weeks of body mass gain, but increased during pre-hibernation fattening. LB juveniles showed higher torpor use, reached similar body sizes but lower fat content than EB individuals before hibernation. Finally, LB individuals showed similar patterns of hibernation, but higher proportion of breeders during the following year than EB dormice. These results suggest that torpor is incompatible with growth but promotes fattening and consolidates pre-hibernation fat depots. In garden dormice, being born late in the reproductive season is associated with a fast life history.
DOI: https://doi.org/10.7554/eLife.31225.001

## Introduction

Because of limited food resources in the environment, organisms have to optimize the allocation of energy to the competing processes of self-maintenance, growth and reproduction to maximize their fitness. In seasonal environments, animals often have to face periods of food shortage and thus trade-offs of energy allocation. In response to these challenges animals may employ diverse strategies to cope with unfavourable conditions. These include migration, energy storage or the use of hypometabolic states. In particular, many small mammals and birds have developed specific mechanisms of energy and water conservation, achieved by active and controlled reductions of metabolic rate ('MR') and hence body temperature ('$T_b$') that is, daily torpor or hibernation (*Geiser and Ruf, 1995*; *Heldmaier and Ruf, 1992*; *Ruf and Geiser, 2015*).

Since most hibernators do not eat during their hibernation season but use body fat reserves as primary fuel, gaining sufficient body fat depots prior to hibernation is an important process for overwintering survival (*Dark, 2005*). This is especially true for juveniles due to their low body size, hence low ability to store fat (*Bertolino et al., 2001*; *Dark, 2005*). As a consequence, pre-hibernation body mass – a proxy for fat mass ('FM') – is a good predictor of winter survival and thus fitness (*Armitage et al., 1976*; *Dark, 2005*; *Lenihan et al., 1996*; *Murie and Boag, 1984*).

Besides fat deposition, body size, and consequently growth, is a crucial determinant of fitness and a major component of life history because it can negatively affect future survival and reproduction, namely the 'silver-spoon effect' (*Arendt, 1997*; *Dmitriew, 2011*; *Metcalfe and Monaghan, 2001, 2003*). Indeed, suboptimal, delayed or even accelerated growth can impair the somatic status and/or the reproductive output, and thus the fitness of individuals (*Metcalfe and Monaghan, 2003*).

*For correspondence:
sylvain.giroud@vetmeduni.ac.at

Competing interests: The authors declare that no competing interests exist.

**eLife digest** Garden dormice are small rodents which are common in European woodlands. They were historically widespread from Portugal in the west to the Urals (Russia) in the east. However they are now largely confined to western Europe with north-eastern and eastern populations having become scattered and fragmented. During the course of a year in northern and central Europe, they make the most of the warm season to fatten up and to produce up to two litters of youngsters. When winter comes, dormice enter hibernation, sometimes for more than six months. During this time, they must rely on their fat reserves to survive. Every year, the young from the second litter have less time to prepare for the winter compared to their siblings born earlier in the season. So, how do they still manage to get ready on time for hibernation?

Here, Mahlert et al. studied captive pups from first and second litters for their first year, following them as they grew up, entered and then emerged from their first hibernation. The late-born individuals developed nearly twice as fast as the ones born early in the season. In fact, both reached a similar body size, but the second-litter dormice had less fat reserves. Just before their first winter, both early- and late-born animals increasingly started to enter torpor – short and daily resting-like periods when the body slows down. Torpor rarely happens when animals are growing (because growth requires a warm body), but it is useful to help storing and consolidating fat before the cold months. Late-born dormice experienced more torpor on average than their first-litter peers.

Both groups survived their first hibernation; but when they emerged, late-born individuals were more likely to reproduce that year. In other words, the dormice which grew quickly might also have sexually matured earlier. This could suggest that animals born later in the season have a faster life history: they grow rapidly, reproduce quickly but may die younger than their early-born peers.

Mahlert et al. highlighted how early-life events can shape the course of animals' existences and influence how their bodies operate. It remains to be examined how these circumstances may affect the individuals in the longer term, and perhaps even their descendants.

DOI: https://doi.org/10.7554/eLife.31225.002

Therefore, investing enough resources in the energy-demanding processes of postnatal growth and pre-hibernation fattening is of high relevance for juvenile hibernators. This can, however, represent a challenge, notably when the time before hibernation onset is shortened, as for juveniles born late in the reproductive season. Further, food availability during key periods of development might also be reduced (*Metcalfe and Monaghan, 2001*), as it can happen late in the reproductive season. Therefore, being a hibernator born late in the summer might trigger coordinated energy-sparing strategies for FM gain at the expense of body growth and subsequent reproductive success.

One of the common responses shown by late-born ('LB') individuals is to grow and gain mass at higher-than-normal rates. This is indeed the case in LB garden dormice (*Eliomys quercinus*) that grow almost twice as fast as early-born ('EB') individuals, reaching lower fat content prior to hibernation (*Stumpfel et al., 2017*). One possible strategy to lower energy requirements and to minimize the impact of energetic challenges is the use of short torpor bouts during certain parts of the day, namely daily torpor (*Ruf et al., 1991*). In particular, torpor use is thought to sustain and promote growth, body mass gain and pre-hibernation fattening and may ensure survival in young individuals (*Geiser and Brigham, 2012*; *Geiser et al., 2006*; *Giroud et al., 2012*; *Giroud et al., 2014*). Previous studies of our group have demonstrated that LB juvenile garden dormice use daily torpor to counteract intermittent periods of fasting, showing similar trajectories of body mass gain as *ad libitum*-fed individuals prior to hibernation (*Giroud et al., 2012*; *Giroud et al., 2014*). However, we presently know very little about the use of daily torpor in juvenile hibernators (*Geiser, 2008*; *Geiser and Brigham, 2012*). Further, to our knowledge no study so far has specifically investigated the effect of torpor and alternative energy-saving strategies on processes of growth and pre-hibernation fattening in hibernators according to their time of birth within the season.

The garden dormouse, henceforth 'dormouse', is a small nocturnal and omnivorous rodent endemic to Europe (*Bertolino et al., 2008*; *Vaterlaus-Schlegel, 1997*; *Vogel, 1997*). The hibernation season, during which they solely rely on body energy reserves accumulated prior to hibernation, can last up to 7 months (*Bertolino et al., 2001*). Because a second litter late in the season is

possible, both in captivity and in the field (*Giroud et al., 2012*; *Giroud et al., 2014*; *Moreno, 1988*), dormice provide an excellent model to determine differences between juveniles from EB and LB litters.

In the present study, we therefore investigated the effects of the time of birth (EB vs. LB) and food availability (*ad-libitum* 'AL' vs. intermittently fasted 'IF') on (i) the rate of growth and pre-hibernation fattening, (ii) the use of energy saving strategies such as daily torpor, (iii) the body size and body fat content prior to hibernation, (iv) the subsequent overwinter hibernating patterns and mass losses, and (v) the fecundity at the following spring of female juvenile dormice. Specifically, we hypothesized that LB juvenile dormice (i) grow and fatten at higher rates, due to (ii) a higher use of torpor, and (iii) reach similar pre-hibernation levels of body size but a lower body fat content, leading to (iv) differences in hibernating patterns and mass losses overwinter, compared to EB individuals. Further, we expected IF juveniles to employ more often and longer torpor bouts than AL fed individuals to reach similar levels of body fat content prior to hibernation. Finally, because of higher growth rates, we expected (v) LB juveniles to show a lower level of fecundity than EB individuals, at the subsequent reproductive season.

## Results

### From the early post weaning phase to the onset of hibernation
#### Structural growth, body mass gain and body composition
At the start of the experiments, LB dormice were smaller despite similar body mass (EB vs. LB: 10.9 ± 0.2 cm vs. 10.4 ± 0.1 cm, $F$ = 5.56, p<0.05). Further, LB dormice grew on average two times faster than EB individuals (EB vs. LB: 0.2 ± 0.01 vs. 0.4 ± 0.01 cm week$^{-1}$; *Table 1* and *Figure 1a*). Thus, LB dormice reached the same body size prior to hibernation (EB vs. LB: 12.8 ± 0.1 vs. 12.5 ± 0.1 cm; *Table 1* and *Figure 1a*) within a shorter period of time (EB vs. LB: 9.5 ± 0.2 vs. 5.0 ± 0.1 weeks) compared to EB litters (*Table 1*).

LB juveniles culminated their body mass gain significantly ~4.5 weeks earlier than EB individuals (EB vs. LB: 9.6 ± 0.2 vs. 5.1 ± 0.1 weeks, see *Table 1* and *Figure 1b*). Although LB dormice gained mass at a significant 1.7-fold faster rate than EB dormice (EB vs. LB: 5.6 ± 0.2 vs. 9.4 ± 0.3 g week$^{-1}$), their pre-hibernation body mass level remained ~13 g lower than the EB group (EB vs. LB: 103.1 ± 2.5 vs. 89.9 ± 2.3 g). We found no effect of fasting (AL vs. IF) on growth and body mass gain (*Table 1*).

At the start of the experiments, that is, early post-weaning, LB individuals were significantly fatter than EB juveniles (4.34 ± 1.60 vs. 2.27 ± 0.39 g, $t$ = 3.09, p<0.05). Prior to hibernation, LB individuals showed lower FM, but also lower FFM levels, compared to EB juveniles (*Tables 2* and *3*). Further, we found no difference of FM and FFM levels between dietary groups in both EB and LB juveniles (*Tables 2* and *3*).

**Table 1.** Parameters of analyses of variance for the effects of group (early-born, late-born) and diet (fed ad libitum or intermittently fasted) on the rate ('slope') of body mass gain and structural growth (assessed by body length), the maximal level ('plateau') at which BM and BL levelled off and the time ('time of plateau') at which the plateau of BM or BL occurred during the pre-hibernation period. p-values shown in bold correspond to statistically significant and interpretable values.

| | | Response variables | | | | | |
| | | Slope | | Plateau | | Time of plateau | |
| Parameter | Term | F-value | p-value | F-value | p-value | F-value | p-value |
| --- | --- | --- | --- | --- | --- | --- | --- |
| Body mass | Group | 70.93 | **<0.001** | 5.73 | **<0.05** | 368.71 | **<0.001** |
| | Diet | 1.00 | 0.33 | 0.05 | 0.83 | 0.07 | 0.80 |
| | Group x Diet | 3.58 | 0.07 | 0.13 | 0.72 | 0.44 | 0.52 |
| Body length | Group | 103.55 | **<0.001** | 1.46 | 0.24 | 190.11 | **<0.001** |
| | Diet | 0.01 | 0.92 | 0.01 | 0.97 | 0.43 | 0.52 |
| | Group x Diet | 1.65 | 0.21 | 0.12 | 0.73 | 0.29 | 0.60 |

DOI: https://doi.org/10.7554/eLife.31225.004

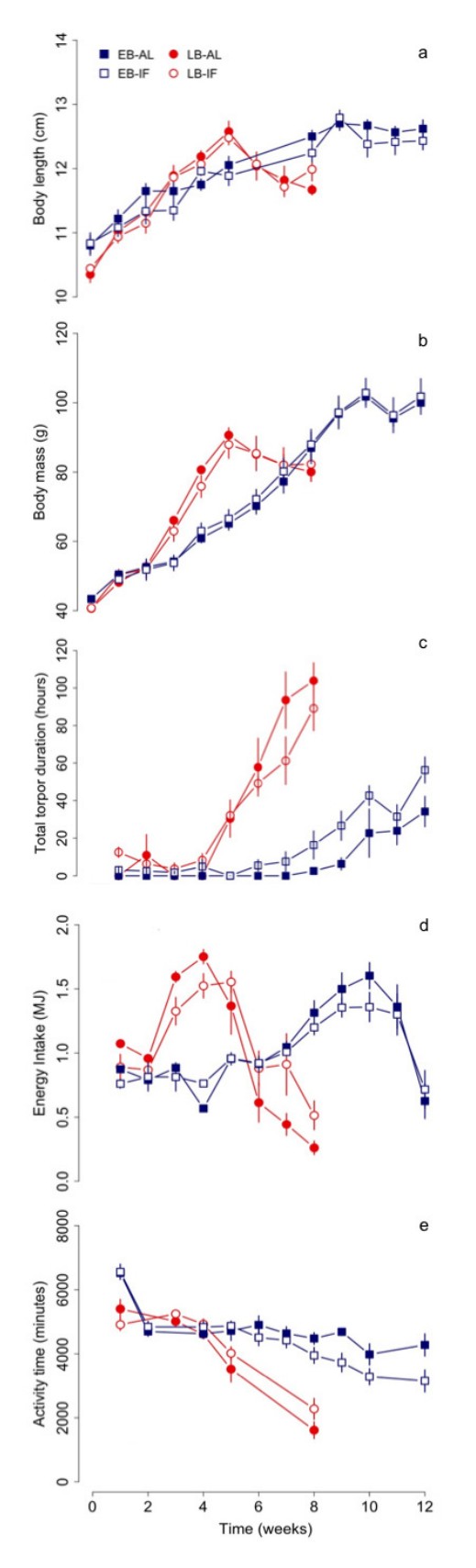

## Total energy expenditure and water turnover at the onset of hibernation

Prior to hibernation, TEE did not significantly differ between EB-AL, EB-IF and LB-AL juveniles, but was significantly higher in LB-IF individuals, compared to the three other groups (*Supplementary file 1*, *Figure 2a*). Prior to hibernation, $rH_2O$ of LB-IF juveniles was also significantly higher than the one of LB-AL and EB-F individuals, although it did not significantly differ from the EB-AL juveniles (*Supplementary file 1*, *Figure 2b*). Further, $rH_2O$ of LB-AL was significantly lower than EB-AL (*Figure 2b*).

## Metabolic rate during the growth and pre-hibernation fattening period

All juvenile garden dormice significantly decreased (irrespective of increased body mass) their ADMR, MR-day and MR-night during the experimental period (*Supplementary file 2*). No significant differences in ADMR were found, neither between EB and LB juveniles nor between IF and AL individuals (*Supplementary file 2*). However, LB dormice had a significantly lower MR-night than EB juveniles (*Supplementary file 2*). Further, IF individuals showed a lower MR-day compared to AL juveniles (*Supplementary file 2*).

## Energy intake

The time course of energy intake differed between EB and LB dormice (*Figure 1d*). LB individuals already started to increase their energy intake after the second week with a peak at week 4, while EB only started to increase their energy consumption from week 6 and reached a peak in the 10th week. LB dormice consumed on average $8.27 \pm 0.42$ MJ over the entire study period (Ecum) and thus significantly less energy than EB individuals ($12.14 \pm 0.25$ MJ, $F = 16.01$, p<0.001). However, the average energy consumption per week ($Ecum_{mean}$) was higher in LB dormice (EB vs. LB: $1.01 \pm 0.02$ vs. $1.03 \pm 0.05$ MJ week$^{-1}$; $F = 14.13$, p<0.01). Energy intake in EB dormice increased during the experimental period ($\chi^2 = 219.81$, p<0.001), irrespective of diet ($\chi^2 = 0.45$, p=0.50). In contrast to EB, the temporal pattern of energy intake differed between the dietary groups in LB animals (diet x time: $\chi^2 = 41.58$, p<0.001). This suggests that LB individuals fed AL had a higher energy consumption before the peak (week 4), but a lower one after the peak compared with IF dormice. In both, EB and LB dormice energy intake was

**Figure 1.** Changes in (a) structural growth (body length), (b) body mass, (c) total torpor duration, (d) energy intake and (e) activity time of early ('EB') and late-born ('LB') female juvenile garden dormice either fed *ad libitum* ('AL') or intermittently fasted ('IF') from the start of experiments until hibernation onset. N = 6 individuals per group. Values are means ±SE. Torpor duration and activity time were estimated from measurements of nest temperature (for details see the 'Material and methods' part).

DOI: https://doi.org/10.7554/eLife.31225.003

positively affected by body mass (EB: $\chi^2$ = 95.55, p<0.001, LB: $\chi^2$ = 46.36, p<0.001).

## Torpor use

LB dormice already started to increase torpor use (both frequency and durations) in week 4 thus significantly earlier than EB individuals, which started to increase their use of torpor not until the eighth week of experiments (*Figure 3*, *Table 4* and *Supplementary file 3*). Hence, both groups started to increase use of torpor 1 to 2 weeks before termination of growth and body mass gain, whereby LB dormice started to increase torpor use at lower levels of body mass (EB vs. LB: 87.5 ± 3.0 g vs. 78.3 ± 1.8 g, F = 6.97, p<0.05) and body size (EB vs. LB: 12.4 ± 0.1 vs. 12.1 ± 0.1, F = 4.34, p<0.05) compared to EB dormice. Further, only IF juveniles of both groups displayed bouts of torpor from the first week (*Figure 3a*). We also found differences in the start of increasing torpor use, that is, starting time, between IF and AL dormice, irrespectively of time of birth (*Table 4* and *Supplementary file 3*). Thus, IF juveniles started to increase their use of torpor on average 1 week earlier compared with AL dormice.

Moreover, LB juveniles increased their mean and total torpor durations to a significant larger extent than EB dormice (*Figure 3b* and *Figure 1c*), but this group effect was dependent on diet (*Table 4*). This significant group-diet interaction also revealed an opposing effect of diet on the increase of total torpor duration within groups. Whereas LB-AL juveniles displayed a significant larger increase of total torpor duration than LB-IF animals, EB-AL individuals increased significantly less their total torpor duration than EB-IF (*Figure 1c*). Further, the slope of torpor frequency was affected by a group-diet interaction (*Table 4*). This suggests that LB-IF dormice increased their torpor frequency by far more rapidly, using torpor most frequently, and displayed shorter mean torpor durations than LB-AL individuals (*Figure 3b*). GLMMs further revealed that although EB-IF dormice displayed higher total torpor duration compared to EB-AL juveniles in some weeks (diet x time: $\chi^2$ = 117.45, p<0.001; *Figure 1c*), the mean torpor duration did not differ significantly (diet: $\chi^2$ = 0.01 p=1.00; *Figure 1c*) because both groups used torpor as frequently (diet: $\chi^2$ = 1.25,

**Table 2.** Parameters of linear models for the effects of time, group (early-born, late-born) and diet (fed ad libitum or intermittently fasted) on body mass, fat mass and fat-free mass of juvenile garden dormice.

p-values shown in bold correspond to statistically significant and interpretable values.

| Response variable | Term | Estimate ± SD | t-value | p-value |
|---|---|---|---|---|
| Body mass (g) | Time | 34.39 ± 2.17 | 15.85 | <0.001 |
| | Group | −9.13 ± 4.36 | −2.09 | 0.06 |
| | Diet | −5.36 ± 4.89 | −1.10 | 0.29 |
| | Time x Group | −16.63 ± 2.71 | −6.14 | **<0.001** |
| | Time x Diet | −0.79 ± 2.73 | 0.29 | 0.78 |
| | Group x Diet | 9.14 ± 6.26 | 1.46 | 0.17 |
| Fat mass (g) | Time | 29.68 ± 1.70 | 17.49 | <0.001 |
| | Group | −3.61 ± 2.80 | −1.29 | 0.22 |
| | Diet | −2.52 ± 3.10 | −0.81 | 0.43 |
| | Time x Group | −13.36 ± 2.11 | −6.32 | **<0.001** |
| | Time x Diet | −2.02 ± 2.13 | −0.95 | 0.36 |
| | Group x Diet | 5.41 ± 3.90 | 1.39 | 0.19 |
| Fat-free mass (g) | Time | 4.72 ± 0.86 | 5.48 | <0.001 |
| | Group | −5.36 ± 1.95 | −2.75 | <0.05 |
| | Diet | −2.82 ± 2.20 | −1.28 | 0.22 |
| | Time x Group | −3.45 ± 1.07 | −3.21 | **<0.01** |
| | Time x Diet | 1.19 ± 1.08 | 1.10 | 0.29 |
| | Group x Diet | 3.76 ± 2.83 | 1.33 | 0.21 |

DOI: https://doi.org/10.7554/eLife.31225.005

**Table 3.** Means and standard deviations for pre- and post-hibernation body mass, fat mass and fat-free mass of the experimental animal groups, according to the time of birth (early-born 'EB', late-born 'LB') and feeding treatment (*ad libitum* 'AL' and intermittently fasted 'IF').
Groups differing significantly (p<0.05, Tukey's post-hoc comparisons) are denoted by different superscripts.

| Variable | Time | Group | Mean ± SD |
|---|---|---|---|
| Body mass (g) | Pre-hibernation | EB-AL | 100.0 ± 0.7[a] |
| | | EB-IF | 101.8 ± 2.1[a] |
| | | LB-AL | 77.4 ± 1.3[b] |
| | | LB-IF | 77.3 ± 2.1[b] |
| | Post-hibernation | EB-AL | 65.7 ± 0.4[c] |
| | | EB-IF | 63.7 ± 1.0[c] |
| | | LB-AL | 59.3 ± 0.6[c] |
| | | LB-IF | 60.5 ± 1.7[c] |
| Fat mass (g) | Pre-hibernation | EB-AL | 47.5 ± 0.6[a] |
| | | EB-IF | 49.2 ± 1.4[a] |
| | | LB-AL | 32.6 ± 1.0[b] |
| | | LB-IF | 32.3 ± 1.3[b] |
| | Post-hibernation | EB-AL | 18.5 ± 0.2[c] |
| | | EB-IF | 18.0 ± 0.7[c] |
| | | LB-AL | 15.5 ± 0.4[c] |
| | | LB-IF | 17.6 ± 0.9[c] |
| Fat-free mass (g) | Pre-hibernation | EB-AL | 52.5 ± 0.4[a] |
| | | EB-IF | 52.6 ± 0.7[a] |
| | | LB-AL | 44.8 ± 0.5[bc] |
| | | LB-IF | 44.9 ± 0.9[bc] |
| | Post-hibernation | EB-AL | 47.2 ± 0.5[bc] |
| | | EB-IF | 45.7 ± 0.4[c] |
| | | LB-AL | 43.9 ± 0.5[c] |
| | | LB-IF | 43.0 ± 4.3[c] |

DOI: https://doi.org/10.7554/eLife.31225.006

p=0.26). As indicated by a significant diet-time interaction (diet x time: $\chi^2$ = 24.82, p<0.001), mean torpor duration within the LB group was higher in IF juveniles in the first four weeks, whereas LB-AL individuals had higher mean torpor duration than IF individuals in the last weeks prior to hibernation (*Figure 3b*).

Energy intake had only a significant effect on torpor use in LB dormice ($\chi^2$ = 1.27, p=0.26 vs. $\chi^2$ = 29.83, p<0.001), suggesting that LB animals with low energy consumption used torpor more often and showed longer torpor bouts, compared with individuals with high food intake.

## Activity time

The activity time of EB and LB juveniles decreased significantly over time (EB: $\chi^2$ = 90.17, p<0.001, LB: $\chi^2$ = 44.28, p<0.001, *Figure 1e*). This was even true when mean torpor duration, which had a significant effect on activity time (EB: $\chi^2$ = 19.05, p<0.001, LB: $\chi^2$ = 8.80, p<0.01), was included as a random factor in the model (EB: $\chi^2$ = 99.01, p<0.001, LB: $\chi^2$ = 136.71, p<0.001). Moreover, the start of decrease in activity time (starting time) occurred concomitantly with the increase of mean torpor duration (*Table 4*).

Further, the diet treatment had a different effect on the start of decrease in activity time between EB and LB individuals, which was reflected by a significant group-diet interaction (*Table 4*). The reduction in activity time both in LB-AL and LB-IF juveniles started approximately from week 3 (LB-AL: 3.5 ± 0.3, LB-IF: 3.3 ± 0.2 weeks), whereas EB-IF dormice started to reduce their activity time later than EB-AL individuals (EB-AL: 5.2 ± 0.4, EB-IF: 6.2 ± 0.4 weeks). Moreover, LB dormice decreased their activity time nearly three times faster compared with EB animals (EB vs. LB: −243.37 ± 54.36 vs. −711.81 ± 51.25 min week$^{-1}$, *Table 4*). In addition, GLMMs revealed that activity time of LB individuals differed between diet treatments (AL vs. IF) across weeks (diet x time: $\chi^2$ = 9.58, p<0.05), whereas diet had no effect on the activity time of EB animals (diet: $\chi^2$ = 0.98, p=0.32; diet x time: $\chi^2$ = 16.30, p=0.06). Also energy intake had only a significant effect on activity time in LB individuals (EB: $\chi^2$ = 0.46, p=0.50, LB: $\chi^2$ = 5.36, p<0.05).

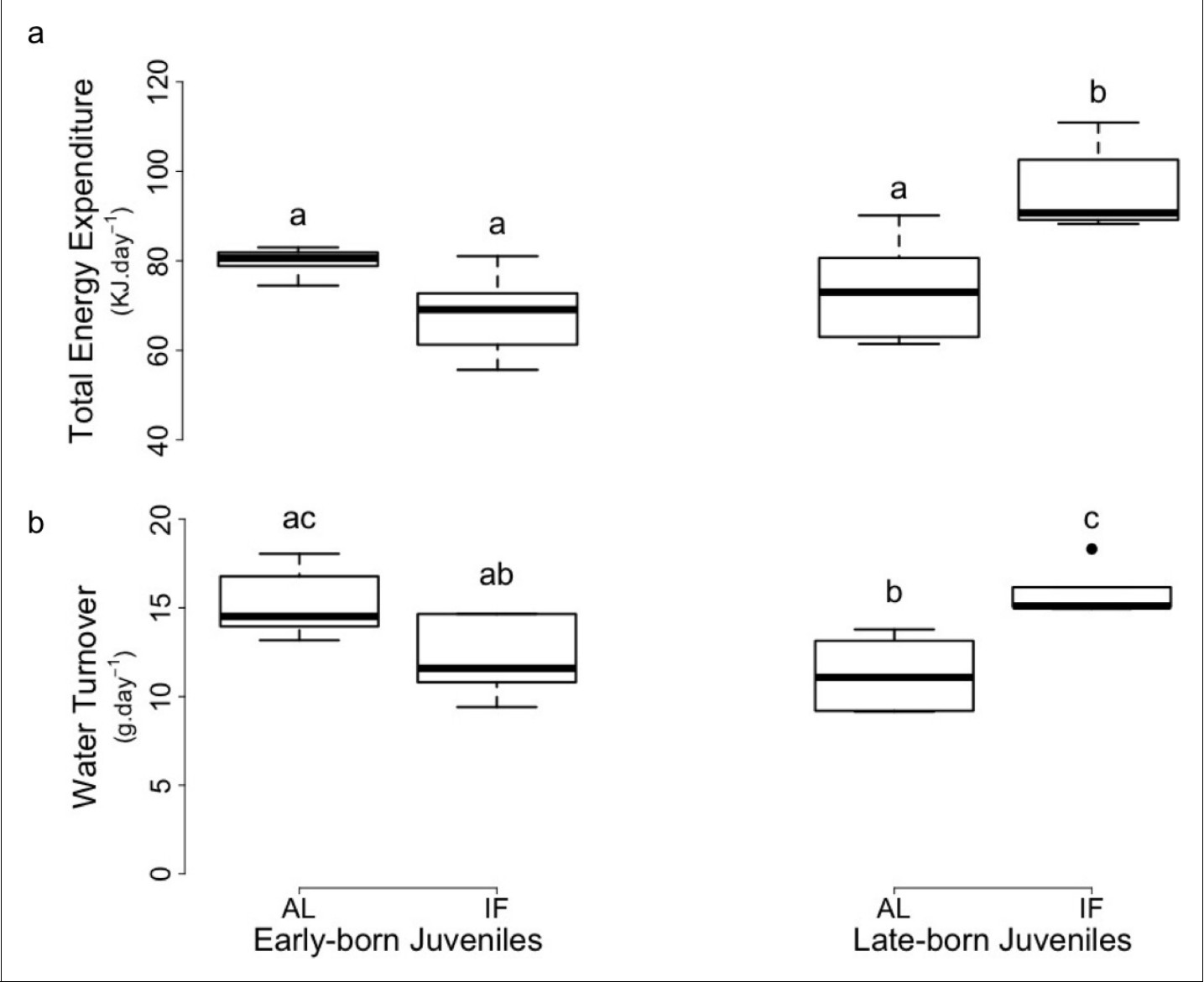

**Figure 2.** Levels of (a) total energy expenditure and (b) water turnover of early and late-born female juvenile garden dormice either fed *ad libitum* ('AL') or intermittently fasted ('IF') prior to hibernation onset (weeks 11–12 for EB and weeks 7–8 for LB). Groups differing significantly (p<0.05, Tukey's post-hoc comparisons) are denoted by different superscripts.

DOI: https://doi.org/10.7554/eLife.31225.007

## Winter hibernation and subsequent reproductive period

### Hibernating patterns

We found a significant group effect on hibernation duration (*Supplementary file 4* and *5*). Indeed, LB juveniles started hibernation 5.5 ± 0.2 weeks later than EB individuals. Both EB and LB individuals terminated hibernation at the same time. The diet treatment had no significant effect on hibernation duration (*Supplementary file 4* and *5*). Further, the arousal frequency, mean torpor and arousal durations, as well as the minimum $T_b$ during torpor did not significantly differ between groups and diets, during the entire winter hibernation (*Supplementary file 4* and *5*).

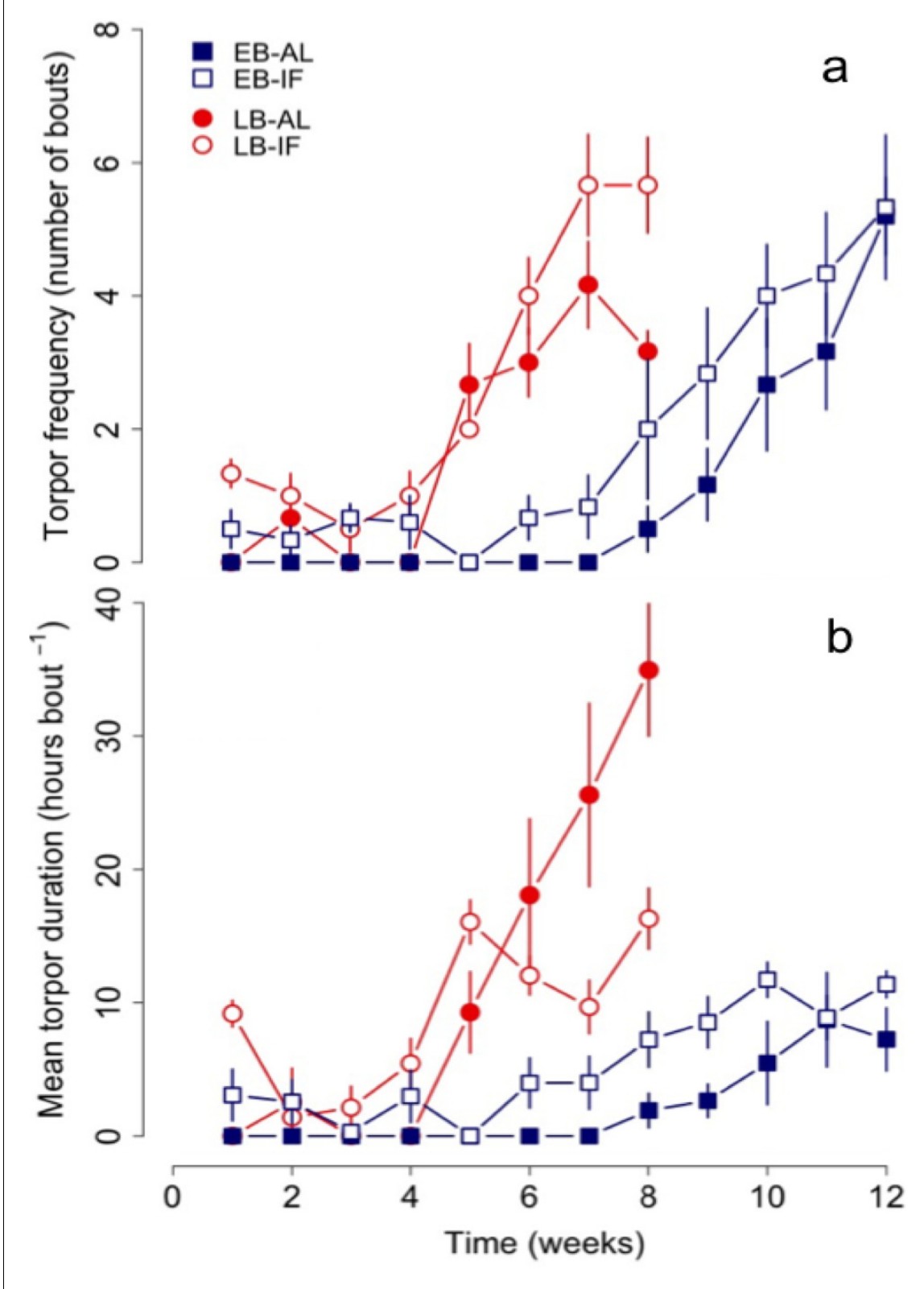

**Figure 3.** Changes in (a) torpor frequency and (b) mean torpor duration of early ('EB') and late-born ('LB') female juvenile garden dormice either fed *ad libitum* ('AL') or intermittently fasted ('IF') from the start of experiments until hibernation onset. N = 6 individuals per group. Values are means ±SE. Torpor patterns were assessed by measurements of nest temperature (for details see the 'Materials and methods' part).
DOI: https://doi.org/10.7554/eLife.31225.008

**Table 4.** Parameters of analyses of variance for the effects of group and diet on the time ('starting time') and the rate ('slope') of torpor increase or decrease of activity time.

Mean torpor duration was also added as a predictor variable in the model for activity time and mean ambient temperature was included as a random effect in all models. p-values shown in bold correspond to statistically significant and interpretable values.

| Response variable | Term | Parameters | | | |
| | | Starting time | | Slope | |
| | | $\chi^2$ | p-value | $\chi^2$ | p-value |
|---|---|---|---|---|---|
| Torpor frequency | Group | 46.47 | **<0.001** | 3.16 | 0.08 |
| | Diet | 8.64 | **<0.01** | 1.18 | 0.28 |
| | Group x Diet | 1.82 | 0.18 | 7.02 | **<0.01** |
| Mean torpor duration | Group | 43.49 | **<0.001** | 38.26 | **<0.001** |
| | Diet | 8.84 | **<0.01** | 0.35 | 0.55 |
| | Group x Diet | 0.50 | 0.48 | 16.64 | **<0.001** |
| Total torpor duration | Group | 43.49 | **<0.001** | 77.86 | <0.001 |
| | Diet | 8.84 | **<0.01** | 0.55 | 0.46 |
| | Group x Diet | 0.50 | 0.48 | 9.66 | **<0.01** |
| Activity time | Group | 0.09 | 0.76 | 6.45 | <0.05 |
| | Diet | 14.06 | **<0.001** | 2.64 | 0.10 |
| | Group x Diet | 6.96 | **<0.01** | 0.22 | 0.64 |
| | Mean torpor duration | 7.61 | **<0.01** | 2.68 | 0.10 |

DOI: https://doi.org/10.7554/eLife.31225.009

## Body mass and body composition during and at emergence from winter hibernation

Body mass, FM and FFM did not significantly differ between animal groups after winter hibernation (*Tables 2* and *3*). EB individuals lost significant higher proportions of body mass and FM compared to LB juveniles over hibernation (*Tables 2* and *3*, *Figure 4*). The 36% and 24% body mass decreases were mainly accounted by 30% and 22% FM losses, in EB and LB juveniles, respectively (*Figure 4*). Further, FFM was significantly reduced by 6.0 ± 2.3% in EB juveniles, whereas no significant FFM loss was detected in LB individuals over hibernation (*Tables 2* and *3*, *Figure 4*).

## Body mass during the subsequent reproductive period

We found significant effects of time and group, but not of diet treatment, on body mass during the subsequent reproductive season (*Supplementary file 6*). Body mass of all individuals significantly increased from post-hibernation to the start of late breeding event (*Supplementary file 7*). At each time-point, that is, post-hibernation, and at the onset of the early breeding and late breeding, body mass did not differ significantly between juvenile dormice. Further, we did not observe any significant difference in changes of body mass between groups and diet treatments from post-hibernation to early breeding, and then until the start of the late breeding event (*Supplementary file 6* and *7*).

## Fecundity

We found significant effects of the timing of reproduction (early vs. late event), the time of birth of the mothers (EB vs. LB) and the dietary treatment (AL vs. IF) on the breeding proportion of the female juvenile dormice (*Figure 5*, *Supplementary file 8*). Overall, females bred 30% less at the late-reproductive event compared to the early-reproductive event. Further, LB females bred 18% more than EB individuals, and IF females bred 16% less than AL animals. We did not find any significant effects of the timing of reproduction, the condition of the mothers and the dietary treatment on the litter size or on litter mass at birth (*Supplementary file 8*).

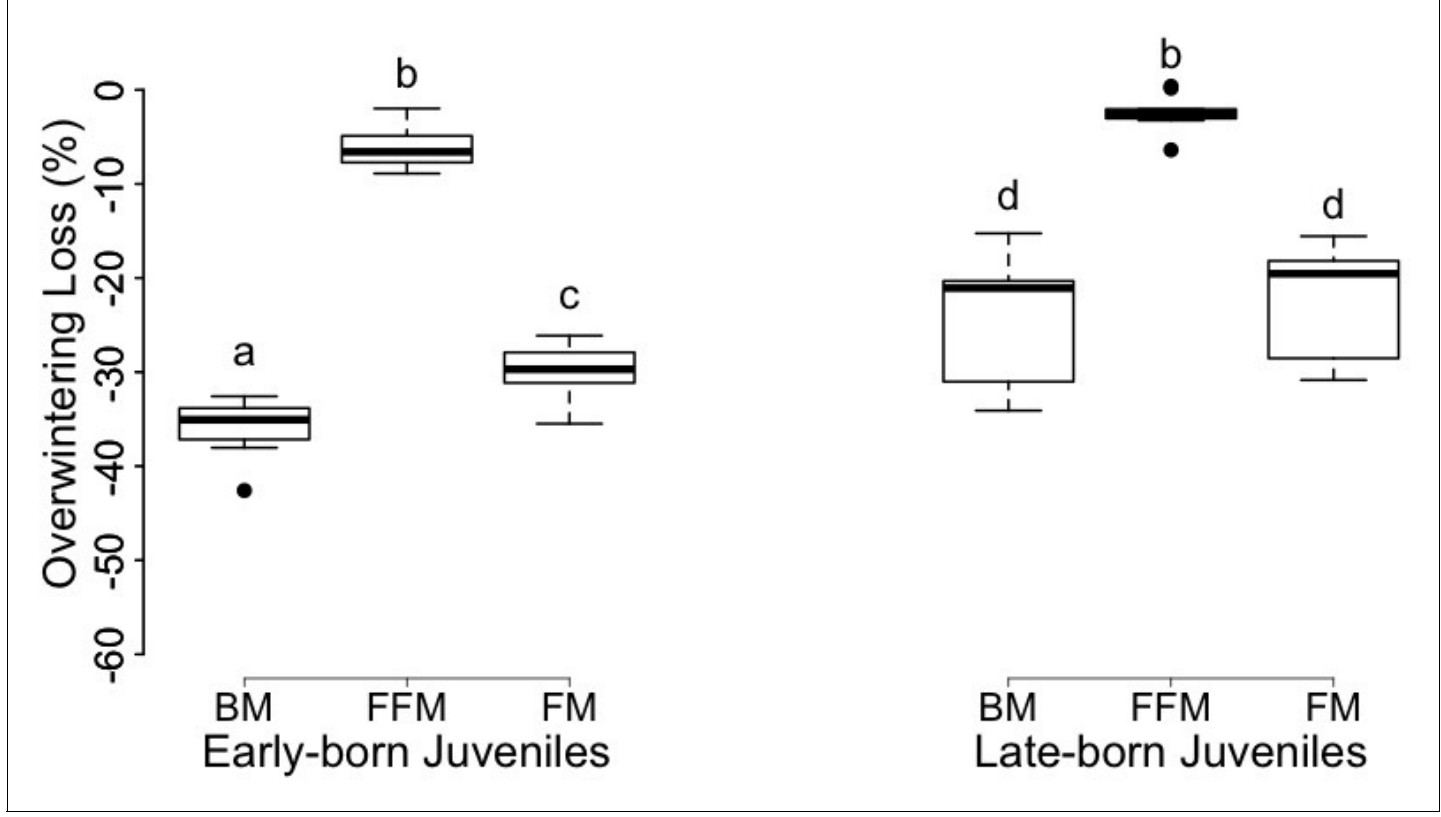

**Figure 4.** Box-plot graphs of overwintering loss of body mass ('BM'), fat-free mass ('FFM') and fat mass ('FM') of female juvenile garden dormice born early ('Early-born Juveniles') or late ('Late-born Juveniles') in the reproductive season. Groups differing significantly (p<0.05, Tukey's post-hoc comparisons) are denoted by different superscripts.
DOI: https://doi.org/10.7554/eLife.31225.010

## Discussion

### Accelerated growth and fattening in late-born juveniles

In line with our previous findings in juvenile garden dormice (*Stumpfel et al., 2017*), our results indicate that LB dormice increased their rates of growth and body mass gain compared to EB individuals. This supports the time-stress hypothesis that accelerated growth is most likely to occur when the time available for reaching a critical developmental milestone is constrained, as for instance when born late in the reproductive season (*Clark, 1970*; *Dmitriew, 2011*; *Lee et al., 2012b*; *Mangel and Munch, 2005*; *Metcalfe et al., 2002*; *Neal, 1965*).

Similar to findings of *Gotthard (2000)* and *Nylin et al. (1996)* that butterfly larvae (*Lasiommata maera* and *Pararge aegeria*) grew significantly faster when maintained under a late-season photoperiod, one might assume that day-length also acts as a crucial environmental cue for dormice to regulate their rate of growth. This might be even true for embryonic and early postnatal development, when offspring itself had no access to daylight, since mothers can transfer photoperiodic information to offspring through the placenta and/or milk via the hormone melatonin (*Bishnupuri and Haldar, 1999*; *Davis, 1997*; *Gündüz and Stetson, 2003*). Thus, melatonin can be the foetus' window to periodicity of the outside world as described by *Davis (1997)* and may prompt LB dormice to accelerate their development. This might explain why LB juvenile dormice gained body mass at a faster rate before the start of the study period, that is, during the maternal phase, and could hence be separated from the mother earlier but with a similar body mass compared with EB individuals.

Further, LB juveniles were structurally smaller but fatter compared to EB individuals at the start of the experiments. Since juveniles were separated from the mothers shortly after the weaning time, feeding on solid food before the separation of LB individuals from the mothers might have had a

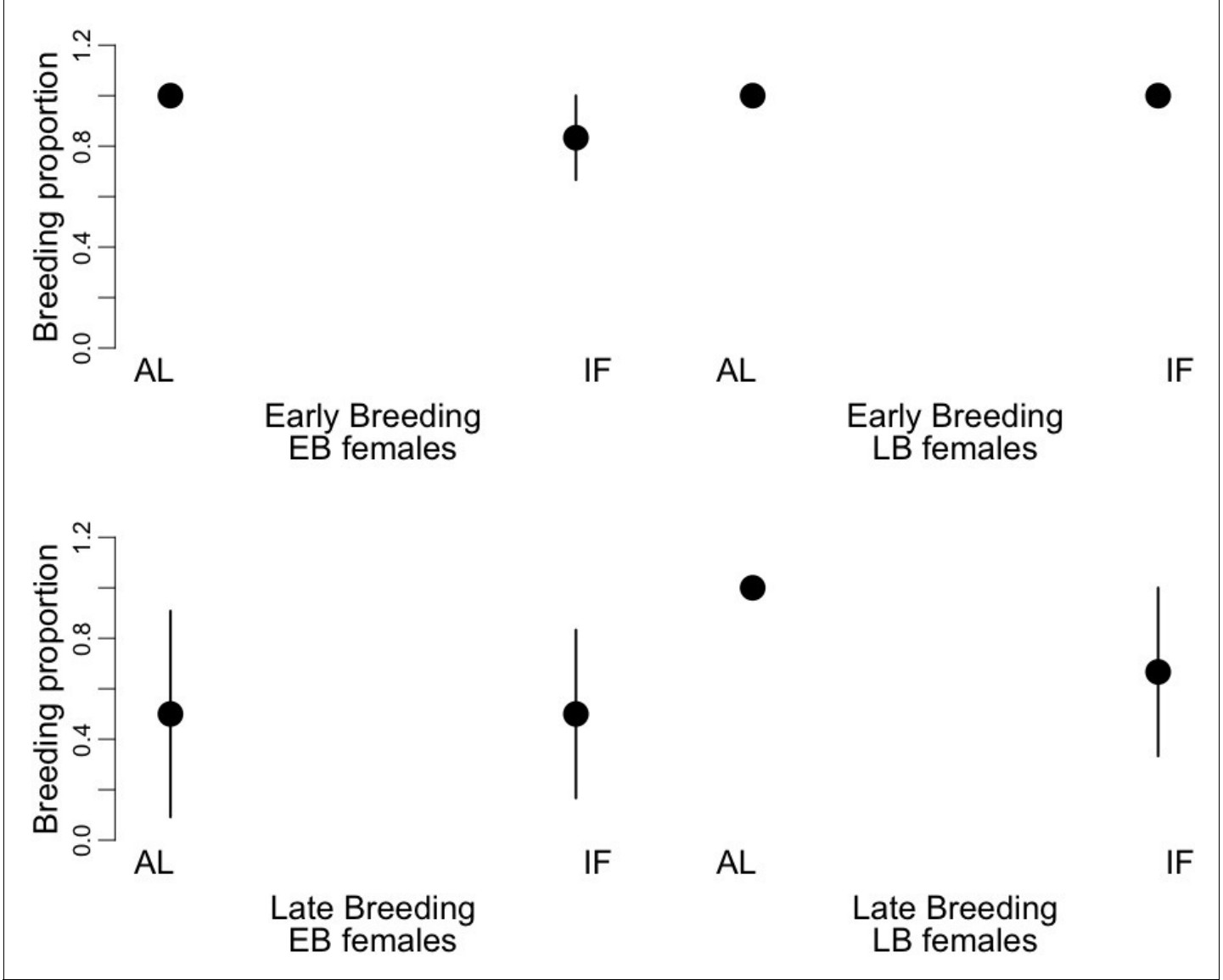

**Figure 5.** Proportion of breeding occurrence ('breeding proportion') of early-born ('EB') and late-born ('LB') female garden dormice at early and late reproductive events. Females were either fed *ad libitum* (AL) or intermittently fasted ('IF') during their juvenile period, that is, from the start of experiments until hibernation onset. Values are means ±SD.

DOI: https://doi.org/10.7554/eLife.31225.011

minor effect on body mass at the start of the study. Hence, it is likely that differences in maternal energy allocation to offspring, for example due to seasonal changes in maternal condition (*Epstein, 1978*; *Skibiel et al., 2009*; *Valencak et al., 2009*), were responsible for the observed two-fold higher amount of FM in LB juveniles compared to EB dormice at the early post-weaning phase. Future studies are clearly needed to determine possible effects of maternal care and photoperiod per se on juvenile growth rates.

By increasing their growth rates and body mass gain, LB dormice were able to catch up in body size but they ended the season with a 16% lower body fat content compared with EB dormice. Our study further revealed that intermittent fasting had no effect on growth rate, maximal pre-hibernation body size and fat content, in either EB or LB individuals. These results are in agreement with previous findings on LB juvenile garden dormice. Indeed, LB dormice that were fasted 3 days a week reached similar body masses compared to individuals fed *ad libitum* (*Giroud et al., 2014*). Also, LB individuals fasted twice a week and experiencing a gradually decreasing $T_a$ gained mass at

an even higher rate than *ad libitum* fed animals (*Giroud et al., 2012*). These results are also in line with previous studies on other hibernating species (*Dark, 1984*; *Dark et al., 1986*; *Hatton et al., 2017*). For instance, arctic ground squirrels that were calorie restricted by 50% during 5 weeks of pre-hibernation fattened at the same rate and reached similar amount of fat reserves to *ad libitum* fed individuals (*Hatton et al., 2017*). Also, golden-mantled ground squirrels with restricted pre-hibernation food intakes underwent similar increases of body mass and possessed equivalent amounts of abdominal white adipose tissue as animals fed *ad-libitum* (*Dark, 1984*).

## Torpor use during growth and pre-hibernation fattening

Dormice in this study differed in their use of torpor depending on the time of birth and food availability. Further, we can distinguish two different phases of torpor use: (i) a low torpor use during the intensive growth phase, that are, the first 6 weeks for EB and the first 4 weeks for LB, and (ii) an increasing torpor use prior to the start of hibernation, that is, after the termination of growth and when the body mass plateaued.

1. Our results suggest that an increased use of torpor seems to conflict with the energy-demanding processes of structural growth. Supporting evidence for this hypothesis is provided by a recent study of *Scherbarth et al. (2015)* showing that a suppression of growth hormone secretion via pasireotide, an analog of the growth hormone release inhibiting hormone somatostatin, increased the incidence of daily torpor in Siberian hamsters (*Phodopus sungorus*) exposed to a (winter-like) short photoperiod. Hence, high levels of growth hormone required for the activation of the growth axis may prevent individuals from entering a torpid state. Indeed, it is well understandable that low $T_b$ during torpor is incompatible with the high metabolic process of structural body growth. The higher levels of MR during the first weeks of the experiments, that is, of intensive growth, compared to those prior to hibernation, that is, of increased torpor use, further support this idea.

2. Female juvenile dormice started to increase their use of torpor during the late phase prior to hibernation, after the completion of growth, while body mass was still increasing, at least in EB individuals. These results (1) confirm previous conclusions (*Giroud et al., 2014*), that dormice increase their use of torpor late in the season to support pre-hibernation fattening rather than growth and (2) are in line with recent findings from female arctic ground squirrels that significantly lowered their mass-specific resting metabolic rate (RMR) between early May and mid-July when animals were completing their mass gain prior to hibernation (*Sheriff et al., 2013*). Female juvenile dormice continued to increase their torpor use (both frequency and durations) while their body mass, hence fat mass, plateaued until hibernation onset. This result is similar to the observations by *Sheriff et al. (2012)* and *Sheriff et al. (2013)* that $T_b$ and mass-specific RMR of arctic ground squirrels continued to decline at stable body mass until hibernation, potentially enabling the maintenance of body fat mass previously deposited.

## Energetic strategies in response to lowered food availability and shortened time prior to hibernation

Only fasted dormice (irrespective of time of birth) used regular torpor during the intensive growth phase, which confirms previous studies (*Giroud et al., 2012*; *Giroud et al., 2014*). Therefore, we can conclude that the use of torpor at this time represents a response to the acute energetic challenge, so called fasting-induced torpor (FIT) (*Diedrich et al., 2015*; *Giroud et al., 2008*; *Vuarin and Henry, 2014*). Thus, the lower food availability triggered the use of torpor, in order to lower the energy expenditure, as supported by a lower metabolic rate during daytime in fasted juveniles, compared with *ad libitum* fed individuals.

The fact that *ad libitum* fed dormice avoided regular use of torpor during the growth phase is also in agreement with the finding of *Humphries et al. (2003a)* that food-storing eastern chipmunks (*Tamias striatus*) minimized torpor expression when food hoards were supplemented. These results strengthen the evidence of torpor-related costs, which were also demonstrated in hibernators that rely on fat stores only (*Bieber et al., 2014*; *Zervanos et al., 2014*). As indicated by our present results one of these costs that has previously not been considered could be the incompatibility of growth processes with the use of torpor.

Torpor represents a flexible tool whose use strongly depends on the individual endogenous and exogenous conditions (*Humphries et al., 2003a*). Since LB dormice had in total 5 weeks less time

for growth and fattening before the beginning of hibernation, they were constrained to reduce energy expenditure more strongly by reducing activity time and using torpor earlier and to a greater extent than EB individuals. Further, in agreement with our hypothesis, our study shows that fasting acted as an additive effect on the time of birth by increasing the need for energy saving strategies such as torpor. Hence, the less-challenged EB-AL individuals increased torpor use last and used torpor least, also indicating an avoidance of potential costs. On the contrary, the most-challenged LB-IF dormice exhibited the most frequent use and earliest increase in torpor. However, LB-IF juveniles spent on average and in total less time in torpor, compared to LB-AL animals that employed less frequent but longer torpor bouts during the pre-hibernation period. This suggests that the unpredictability of food availability constrained LB-IF dormice to arouse more often to enable continuation of feeding. This strategy of more frequent arousals led to significant higher total energy expenditure and water turnover in LB-IF individuals, which was compensated by high food intake during days when food was available in order to cover their increased energy needs.

## Energetic strategies and their limits

In general, organisms can increase the amount of energy available for growth and fattening by reducing their energy expenditure and/or by increasing their energy intake. Our data show that LB dormice used a combination of both. Concurrently with the beginning of growth acceleration in week 3, activity time decreased and energy intake increased drastically. These results are supported by the occurrence of a lower MR, during the night, and a higher average food intake per week, in LB juveniles than in EB individuals. Commonly in the wild, a higher food intake is associated with a longer time that must be spent for foraging outside the nest, thus a higher activity time (e.g. *Bieber et al., 2017*). Therefore it seems likely that LB juveniles benefited from captive conditions since foraging was associated with low effort unlike in the wild. Despite an increased intake of food, LB dormice did not reach pre-hibernation body FM levels similar to EB individuals. Thus, it seems that LB dormice might have exploited the full potential of this compensatory strategy. Energy consumption can be limited by stomach volume as well as the trade-off between retention time and assimilation efficiency. The ingestion of higher amounts of food per time unit is usually correlated with an accelerated passage rate through the gastrointestinal tract, and hence could decrease digestive efficiency (*Clauss et al., 2007*). On the other hand, torpor may increase energy conversion efficiency by increasing retention time (*Humphries et al., 2001*), although digestive activity at low $T_b$ remains debatable. So whether the strategy of higher energy consumption was indeed stretched to its limit or if the costs of even faster fat accumulation and/or growth due to additionally increased ingestion were constraining remain to be determined.

## Consequences for winter hibernation and substrate utilization over hibernation in juveniles

Several studies observed that pre-hibernating body mass, as a proxy of body fat reserves, affects overwinter survival of juvenile hibernators (*Armitage et al., 1976*; *Lenihan et al., 1996*; *Murie and Boag, 1984*). Animals with body masses below a certain threshold, as it can be the case for juveniles, are less likely to survive the winter (*Rieger, 1996*). In our study, LB juveniles had lower fat reserves compared to EB individuals at hibernation start. Despite this fact, LB juveniles showed hibernating patterns and survival rates similar to EB individuals. This result is in line with findings on common dormice, from a population also producing two litters per year, and in which no significant differences in winter survival between EB and LB juveniles were detected (*Bieber et al., 2012*). The pre-hibernation fat reserves of LB juveniles in our study were hence sufficient to fuel their shorter winter duration due to the five-weeks delay of hibernation start.

Interestingly, we found that the overwintering body mass loss of juveniles was mainly accounted for a reduction of FM. This result contrasts with our previous findings on equal reductions of FM and FFM in garden dormice during winter hibernation (see *Figure 3*, *Giroud et al., 2014*). However, in *Giroud et al. (2014)*, we determined pre-hibernation BC of the animals at the end of the period of highest summer mass gain, at a ~10% lower body (fat) mass compared to the peak body mass prior to hibernation. As the last phase of body mass gain prior to hibernation mainly corresponds to fat deposition, pre-hibernation level of FM, but not FFM, might have been largely underestimated. In contrast, in the present study BC was assessed slightly after the peak of body mass prior to

hibernation, closely reflecting BC of the animals at the onset of hibernation. Therefore, the current data provides a rigorous determination of the source (i.e. FM and FFM) of body mass loss overwinter in garden dormice, which almost exclusively point to FM loss during winter hibernation. It is indeed a characteristic of most hibernating species to fuel their energy needs in torpor almost exclusively with fat oxidation, as supported by studies that measured MR and substrate-type utilization during torpor in hibernators kept at $T_a$ between 0°C and 8°C in winter (for review see *Dark, 2005*; *Heldmaier et al., 2004*). However, there is evidence from an extreme hibernator, the arctic ground squirrel (*Spermophilus parryii*), for mixed fuel (lipid/protein) metabolism occurring when animals are exposed to sub-zero $T_a$ (0°C to −16°C; see *Figure 1* in *Buck and Barnes, 2000*). In our study, dormice were hibernating at $T_a$ of 3.8°C to 7°C during five to six months in winter. Hence juvenile dormice were likely oxidizing preferentially lipids over proteins during hibernation, which is in line with the limited loss of FFM (<8%) observed in these animals. Such decrease of FFM can likely be explained by reductions of the gastrointestinal mass and alimentary organs, as previously reported in several hibernators during long-term fasts (*Bieber et al., 2011*; *Carey, 1990*; *Hume et al., 2002*). Nevertheless, we cannot exclude the possibility that dormice also reduced their muscle mass during winter hibernation. For instance, arctic ground squirrels do lose 20% of their lean mass over winter (*Buck and Barnes, 1999*), and show significant remodeling of some essential organs across the hibernation season (*Lee et al., 2012b*). On the other hand, there is evidence for muscle sparing in some hibernating species. For instance, hibernating bears conserve their lean mass and muscle strength during several months of fast in winter (*Lohuis et al., 2007*). Similarly, ground squirrels kept under a very-low-calorie intake are able to fully spare their proteins. Provided that they have access to some food, they manage to compensate for protein utilization, while at the same time selectively losing large amounts of fat (*Karmann et al., 1994*). To date, the conditions and mechanisms for the maintenance of (lean) muscle mass in hibernating mammals over winter are still unclear. The garden dormouse seems to spare its lean mass over winter and therefore appears to also be a good model to study mechanisms of protein sparing during hibernation.

## Consequences for reproductive success on the following year

Because of higher growth rates, we expected LB individuals to show reduced reproductive output during the following year, namely the 'silver-spoon effect'. Indeed, in the past decades various fitness consequences of accelerated growth, based on resource allocation trade-offs between growth, maintenance and reproduction, were reported (*Lee et al., 2012b*; *Lindström et al., 2005*; *Metcalfe and Monaghan, 2001*). For instance, accelerated rates of growth are expected to reduce overall reproductive output and/or lifespan of the individuals (*Metcalfe and Monaghan, 2001*).

Our results are, however, in contradiction with this view, since we found that LB individuals bred at higher proportions than EB animals during the following reproductive year. These findings are in line with the 'internal predictive adaptive response' hypothesis, which predicts that individuals born in poor conditions, such as LB individuals of this study, should start to reproduce earlier if they are likely to have reduced performance later in life. This suggests that being born late in the reproductive season is associated with a fast life history, that is, fast growth, fast reproduction and short lifespan. Although fast and slow life histories of animals were often shown to differ interspecifically, there are also reports on intraspecific differences in life history traits, such as patterns of growth, reproduction and survival. Indeed, slow and fast life cycles were reported in six different populations of Columbian ground squirrels, suggesting that life cycles are phenotypically plastic (*Dobson and Murie, 1987*; *Dobson and Oli, 2007*). Also, wild boars switch from a relatively slow life history under conditions of high food supply to a fast life history when food is scarce (*Bieber and Ruf, 2005*). Hence, growing at faster rate because of a late start in the reproductive season may accelerate the individual pace of life, explaining higher reproductive output during the first season. However, the consequence of a higher reproductive investment during the first year on future reproduction and lifespan remains to be determined. This strategy employed by LB individuals might further reduce their lifespan, hence their breeding opportunities, leading to overall reproductive output similar to, or even lower than EB individuals.

Our results further demonstrate that intermittently fasted juveniles reproduced less during the following year than those that were fed *ad-libitum* prior to hibernation. This finding suggests that food availability during the developmental phase of an individual affects its future reproductive success. Similarly, it seems that past experiences (reproduction and food availability) also affect subsequent

reproductive output in two species of frogs, *Rana esculenta* and *Rana lessonae* (**Waelti and Reyer, 2007**). Indeed, past reproduction negatively affected growth during summer and body condition during autumn, which in turn, reduced the reproductive output at the following year. Further, this reduction of future reproduction was more pronounced under low food availability compared to a high food treatment, in both frog species. Such effects of food availability on future reproduction were also described in mammals, including humans (**Lummaa and Clutton-Brock, 2002**). For instance, restricting food given to golden hamsters (*Mesocricetus auratus*) females during their first 50 days of life causes them to produce smaller litters and female-biased sex ratios during adult life, even after they have been replaced on *ad libitum* diets (**Huck et al., 1986**). In humans, mothers who were exposed to a famine in the Netherlands during the winter of 1944–1945 had reduced weight gain during pregnancy and, consequently, produced infants with reduced foetal growth, birth weight and length, and neonatal survival (**Stein et al., 1995**). In our study, the intermittent fasting treatment during early life might have been interpreted by the juveniles as a signal for an environment with low and unpredictable food availability, leading them to dampen their subsequent reproductive investment, probably to the advantage of their somatic maintenance, hence their fitness.

## Conclusion

Taken together, the present study shows that growth acceleration, enabled by increased food intake and reduced activity duration, acted as compensatory strategy for a late start. This led to pre-hibernation body sizes in LB juveniles that are indistinguishable from those of EB dormice. Further, an extensive use of torpor seemed to be incompatible with structural growth, but supports pre-hibernation fattening late in the season and the preservation of the fat reserves previously deposited prior to hibernation. This was especially the case for EB juveniles that reached higher levels of fat reserves prior to hibernation, compared to LB individuals. LB juveniles delayed their onset of hibernation by 5 weeks. Further, their lower body FM had no consequences for subsequent winter hibernation and survival, showing similar hibernating patterns to EB individuals. For both EB and LB juveniles, overwintering body mass loss was mainly explained by a reduction of FM, meaning that garden dormice spare their lean mass during winter hibernation. Finally, it seems that the fact of being born late in the reproductive season accelerates the individual pace of life, by accelerating growth rate and increasing fecundity in sub-adults during the following year. However, the long-term impact of a late birth on overall reproductive output and lifespan, hence individual fitness, remains to be determined.

## Materials and methods

### Animals

In total, we investigated 36 female juvenile garden dormice from a breeding colony kept at the Research Institute of Wildlife Ecology (University of Veterinary Medicine, Vienna, Austria; latitude 48°15'N, longitude 16°22'E). Animals were reared in an unheated and poorly insulated housing facility under natural variations of photoperiod and ambient temperature ($T_a$). After separation from the mother, juveniles were individually marked with miniature subcutaneous PIT tags (Tierchip Dasmann, ANIMAL ID ISO 11784/85 FDX-B Standard, Tecklenburg, Germany, http://tierchip.de) and were housed separately in cages (60 × 40 × 40 cm). Cages were equipped with branches and self-made plastic nest boxes filled with hay.

### Protocol overview

The experiments involved 18 EB (born from May 17th to May 24th, 2013) and 18 LB (born from Aug 7th to Aug 9th, 2013) female dormice: 12 EB and 12 LB individuals were studied during the pre-hibernation, hibernation and reproductive periods (see below for further details), and six additional EB and six additional LB individuals were sacrificed at the early post-weaning phase for determination of body composition ('BC', see below for further details), in order to minimize impacts on the 24 EB and LB individuals included in the follow-up experiments. The 12 EB or 12 LB and 6 EB or 6 LB juvenile females were selected based on their body mass, at the day of separation from the mothers, that is, at the early post-weaning phase, in order to match the averaged litter mass. The EB individuals were separated from the mothers at an age of 44.0 ± 1.5 days with a body mass of 40.2 ± 0.6 g.

To reduce genetic variability we bred the same parents as for EB individuals a second time in the year to obtain LB litters. Because we assumed the same developmental stage, LB dormice were separated from the mothers at an age corresponding to a body mass similar to the one of EB dormice at their time of separation ($t = 1.40$, p=0.18). Thus, LB juveniles were separated from the mothers at an age of $36.0 \pm 0.7$ days and a body mass of $38.8 \pm 0.7$ g. After the separation from the mothers, the 12 EB and 12 LB animals entered the experiments after 4 days of habituation to cages and nest boxes.

During the pre-hibernation period, over the subsequent winter hibernation and during the reproductive period on the following year, 12 EB and 12 LB juvenile dormice were studied. The pre-hibernation period started from the separation from the mothers until the start of hibernation (EB: 12 weeks from July to mid-October; LB: 8 weeks from mid-September to end of November). The beginning of hibernation was marked by the occurrence of prolonged ($\geq 24$ hr) torpor bouts and the plateauing of body mass. Half of these 12 EB and 12 LB animals were fed *ad libitum* and the second half was intermittently fasted. In the IF group, food was removed for 24 hr during 2 non-consecutive days randomly each week. $T_a$ was recorded at 30 min intervals during the entire experimental period using temperature loggers (EL-USB-2, Lascar Electronics, UK; resolution: 0.58°C, accuracy: $\pm$ 0.58°C). $T_a$ fluctuated between 15.5°C and 29.5°C (mean $T_a$ $\pm$SD: $22.3 \pm 3.1$°C) and between 13.5°C and 24.5°C (mean $T_a$ $\pm$SD: $18.5 \pm 1.9$°C) over the study periods of EB and LB dormice, respectively. Body size, body mass and food intake were measured once a week (see below for further details). During the pre-hibernation phase, torpor use and the time an individual spent outside the nest (henceforth 'activity time') were monitored continuously via nest temperature recordings (see below for further details). Oxygen consumption ('VO$_2$'), that is, MR, was measured at two time-points within the experiments, that is, at the start and at the end of the experimental period (see below for further details).

Just prior to hibernation, all juveniles were implanted with temperature loggers (see below for further details). After surgery, animals were allowed to recover for 8–10 days before hibernation was induced by housing pairs of individuals (2 EB-AL or 2 EB-IF or 2 LB-AL or 2 LB-IF) in two identical ventilated refrigerators (Liebherr GKv 5730), maintained at $5.3 \pm 0.9$°C SD (upper part: $5.5 \pm 1.7$°C CI; lower part: $5.0 \pm 2.0$°C CI), under constant darkness, without food and water. In each refrigerator (one for EB, one for LB juveniles), pairs of AL or IF animals were housed symmetrically on a vertical axis, so that animals from each group experienced very similar temperature variations. All dormice entered hibernation within the first 24 hr after induction. At emergence from hibernation (April), temperature loggers were surgically removed from all animals. Further, BC, total energy expenditure ('TEE') and water turnover ('rH$_2$O') were assessed, by doubly labeled water ('DLW'), in all animals prior to and at emergence from hibernation, before animals were re-fed after several months in hibernation at a temperature of $5.3 \pm 0.9$°C (see below for further details).

At emergence from hibernation and after removing the loggers, body mass was measured and the fecundity was assessed in all females (n = 24) in May for the early reproductive event, and in half of them (n = 12) in August for the late reproductive event (see below for further details and justifications).

## Growth and body mass gain

Since in juvenile dormice, pre-hibernation body mass is highly correlated with body FM (FM as a function of body mass prior to hibernation: intercept = −70.86, slope = $1.20 \pm 0.09$, adjusted $R^2$ = 0.45, n = 16, p=0.005) (*Stumpfel et al., 2017*), we used body mass gain as proxy for fattening. Body mass and structural growth (body length) of each animal were assessed once a week, as previously described in *Stumpfel et al. (2017)*. To minimize measurement errors, we conducted three non-consecutive measurements of body length per individual and used the mean as a body length measurement. In case of a difference $\geq 5$ mm between two identical measurements and the third value we decided to exclude this latter value as an outlier. Further, we included only body length measurements in data analyses that were carried out by one and the same person to improve accuracy of measurements. Therefore, body length data of EB individuals of weeks 6 and 7 were excluded. The apparent decrease in body size after termination of growth can be explained by measurement difficulties resulting from the simultaneous increase in torpor use and the associated body rigidity in torpor. Nevertheless, we can assume that animals indeed terminated their body growth since body mass culminated at the same time.

## Energy intake

Food was provided in form of apple (only during respirometry measurements) and cat food pellets which contained 40% fibre/nitrogen-free extract ('NFE'), 30% protein, 10% fat, 9% water, sufficient minerals and vitamins (Topix, Saturn Petcare GmbH, Bremen, Germany). The nutrient composition that includes the hydration coefficient of the cat food was provided by the manufacturing company. In addition, the water content of the cat food was verified by weighing the dried samples. The energy content of fed apples was provided by the Agricultural Research Service of the United States Department of Agriculture (http://ndb.nal.usda.gov/ndb). Water was provided *ad libitum*. Food intake was measured on a weekly basis by weighing the amount of food provided to each dormouse and the food left in the cages (including spillage) to the nearest gram (PM34 Delta Range, Mettler Toledo, Greifensee, Switzerland). Thereafter, the weekly food intake was calculated as difference between the initial and the remaining food masses, accounting for losses of water content (in case of apples). In some cases, the determination of food consumption was just approximately possible due to spilled drinking water leading to a nearly inseparable mixture of food and faeces. The amount of consumed energy was computed using the caloric values given in *Livesey (1984)* and *Livesey and Elia, 1988*. Accordingly, the gross energy content of protein, fat and fibre/NFE was 23.3 kJ g$^{-1}$, 39.6 kJ g$^{-1}$ and 17.5 kJ g$^{-1}$, respectively. Gross energy intake was calculated from the amount of food consumed per week in grams multiplied by its energy content.

## Torpor use and activity time

During the pre-hibernation period, we used measurements of nest temperature ('T$_{nest}$') as a proxy for T$_b$ to estimate torpor use and activity time of each dormouse, as described by *Willis et al. (2005)* and used in previous studies in juvenile garden dormice (*Giroud et al., 2012*; *Giroud et al., 2014*). T$_{nest}$ was recorded at 1 min intervals and stored data were downloaded once a week.

From the nest temperature recordings, the following parameters were calculated: torpor frequency (number of torpor bouts displayed by an individual per week), total torpor duration (total time an individual spent in torpor per week), mean torpor duration (averaged torpor bout duration an individual displayed per week) and activity time (total time an individual spent outside the nest). For these calculations, we visually identified starting and ending points of torpor bouts and computed thereafter automatically these four parameters by use of an R script. This script analyzed the activity time by calculating the difference between T$_{nest}$ and T$_a$ and building a moving average over 100 values for smoothing. The mean over all values functioned as threshold: temperatures higher than the mean were interpreted as resting in the nest and values less than the mean were interpreted as being active outside the nest. Thus, we distinguished between nocturnal activity and diurnal rest. To also detect cases of short nest leaves by day or returns into the nest during the night, whereby T$_{nest}$ just changes slightly for a short time, we defined an additional threshold of ±0.5°C. Thus, an increase in temperature greater than 0.5°C during nocturnal activity time was interpreted as a return into the nest during night, therefore as inactivity, whereas a decrease greater than 0.5°C during the day was interpreted as leaving of the nest, hence as activity time. The visual distinction between a decreasing T$_{nest}$ due to torpor and a decrease due to a nest-leaving event was possible because T$_{nest}$ decreased much faster while nest leaving compared with torpor onset (see Figure S2 in *Giroud et al., 2014*).

Sample size was reduced in some weeks due to battery failures of data loggers. The activity data were not available for weeks in which we conducted respirometry measurements. However, torpor use was detected during this time by recording a substantial reduction in MR. This allowed us to complement the torpor data obtained with the T$_{nest}$ measurements for the corresponding week.

## Metabolic rate

Respirometry measurements were carried out at two time-points within the experiments: during the first phase of growth, that is, at weeks 2–3 for both EB and LB juveniles, and at the plateau of body mass, that is, at weeks 11 for EB juveniles and 6–7 for LB individuals, prior to hibernation. At each time-point, rates of VO$_2$, that is, MR, were determined, by using an open-flow respirometry system, as previously described in *Ruf and Grafl (2010)*. MR of each dormouse was measured for 48 hr at T$_a$ of 20°C under naturally adjusted photoperiod. Data of the first ~24 hr were considered as habituation time, hence excluded from the analyses. After the first ~24 hr, measurements were interrupted

for ~1 hr to clean cages, to restock or remove food and water, and to weigh the animals. Food was provided in form of apple, whereby the IF group obtained no food at the second day of respirometry measurements (according to their schedule of fasting). Water was provided in form of agarose gel. Because body mass of juveniles changed during the study period, we used a flow rate of approximately 40 l h$^{-1}$ at the beginning of the growing phase and 60 l h$^{-1}$ prior to hibernation. Oxygen concentration was measured using paramagnetic gas analyzer (Servopro 4100, Servomex, Crowborough, UK). For each dormouse, VO$_2$ was calculated in ml O$_2$ h$^{-1}$ on the basis of the flow rate (converted to STPD conditions) using the equation VO$_2$ = FD*(FIO$_2$-FEO$_2$)/(1-FIO$_2$*(1-RQ)) (l/h; FD = dry flow, FIO$_2$ = fractional concentration of O$_2$ in the incoming airflow, FEO$_2$ = fractional concentration of O$_2$ in the outgoing airflow, RQ = respiratory quotient) (*Lighton, 2008*), assuming a RQ of 0.85 for animals fed AL and of 0.70 for IF animals. From these data, we calculated for each animal the following MR variables: average daily MR over 24 hr ('ADMR'), average MR during day ('MR-day') and average MR during night ('MR-night').

## Determination of body composition, total energy expenditure and water turnover

At the early post-weaning phase, BC, that is, fat-free mass ('FFM') and FM levels, was determined by isotopic dilution in six EB and six LB juveniles (body mass: 41.3 ± 2.1 vs. 41.9 ± 2.7 g in EB and LB individuals, respectively). A premixed 5 g/kg dose of DLW, thinned with 3% NaCl to physiological osmolarity, was injected intra-peritoneously into each animal. Isotopic equilibration in body water was determined from an intra-cardiac blood sample collected at 1 hr post-dose under anesthesia. The blood was collected in micro-capillary tubes, which were immediately flame-sealed. Anesthesia was induced by subcutaneous injection of 50 mg kg$^{-1}$ ketamine (Ketamidor10%, Richter Pharma, Wels, Austria) and 8 mg kg$^{-1}$ xylazine (Rompun2%, Bayer, Leverkusen, Germany). Thereafter, euthanasia was administered by intra-cardiac injection of 0.5 ml T61 (T61 ad us. vet, injection solution, Intervet GesmbH, Vienna, Austria).

Pre- and post-hibernation FFM and FM levels, TEE and rH$_2$O were determined, by the multi-point DLW methodology during a 6-day period, after the first 12 or 8 weeks of the pre-hibernation period, that is, at the plateauing of body mass, for EB and LB juveniles, respectively, or at emergence from hibernation for both EB and LB animals, as already used in garden dormice previously (for details on the methodology, see *Giroud et al., 2014*). Analyses were performed by isotope ratio mass spectrometry, at the Department of Ecology, Physiology and Ethology (IPHC, CNRS-UdS, Strasbourg, France), as described previously (*Chery et al., 2015*). The total body water was derived by the principle of isotopic dilution. FFM was calculated assuming a hydration coefficient of 73.2%, which has been reported to be very stable across species (*Schoeller, 1996*), and FM was computed by difference with body mass.

## Implantations of temperature loggers

Just prior to hibernation, all juveniles (n = 24) were implanted with wax-coated temperature loggers (iButtons DS1922L, Dallas/Maxim, Maxim Integrated, San Jose, CA; resolution 0.5°C). All loggers were calibrated in a water bath between 2°C and 40°C. Surgical anesthesia was induced by subcutaneous injection of 50 mg kg$^{-1}$ ketamine (Ketamidor10%, Richter Pharma, Wels, Austria) and 8 mg kg$^{-1}$ xylazine (Rompun2%, Bayer, Leverkusen, Germany). Animals were then maintained under 1.5% isoflurane in an oxygen stream via facemask. For post-operative analgesia 5 mg kg$^{-1}$ ketoprofen (Romefen 10% Merial S.A.S., Toulouse, France) was administered subcutaneously. The operation field was prepared according to standard surgical procedures and covered by sterile surgical drapes. Animals were placed in dorsal recumbency and the abdominal cavity was opened through a 1 cm incision in the *linea alba* to introduce the temperature logger within the abdomen. Peritoneum and abdominal muscles were sutured using synthetic absorbable surgical suture material USP 3/0 (Surgicryl PGA, SMI AG, Hünningen, Belgium) using a single-button suture technique. In addition synthetic absorbable surgical suture material USP 4/0 (Surgicryl PGA, SMI AG, Hünningen, Belgium) was used to suture the skin with an intra-cutaneous suture technique. During the entire procedure, vital parameters [respiration rate, peripheral haemoglobin oxygen saturation as measured by pulse oximetry (SpO$_2$), heart rate] were monitored. All loggers were removed from the animals after hibernation (in April), by following the same surgical procedure.

From the temperature recording of winter hibernation, we computed the hibernation duration, mean inter-bout euthermy ('IBE') duration, mean torpor bout duration and the minimal $T_b$ in torpor.

## Reproduction and fecundity

We assessed the fecundity of EB and LB females during both early (May) and late (August) reproductive events during the following year. All 24 females were included in the early breeding event, but only half of them (three individuals from each experimental group) were used for the late breeding event, due to logistic limitations. At each reproductive event, breeding occurs in outdoor enclosures, in which 3 females (2 AL and 1 IF, or 1 AL and 2 IF) and 1 male were housed together for mating. All males were 2 years old (all EB) and had an average body mass of 152.7 ± 28.6 g at the start of the early reproductive event and of 117.6 ± 6.0 g at the start of the late reproductive event. All animals received *ad-libitum* access to food and water. At each reproductive event, females were first placed in outdoor enclosures without males for habituation. Males were then introduced 3days later. After 4 weeks kept together, that is, the time for habituation and gestation, females were separated from the males and caged individually in rooms under natural fluctuations of $T_a$ and photoperiod, in which they eventually gave birth, if pregnant. The breeding proportion was determined as the proportion of all females included in the mating. Then, litter size and litter mass at birth were assessed for each female.

## Statistical analyses

Statistical analyses were performed using the program R 3.1.0 (*R Development Core Team, 2014*). All models were assessed visually (via histograms and QQ-plots) for normality of residuals. If necessary, response variables were Box-Cox transformed to achieve a normal distribution. Results of the models are given in terms of $F$ or $\chi^2$ and $p$ values from type-II analysis of variance ('Anova' in the R package car *Fox and Weisberg, 2014*). If not stated otherwise, data are given as mean ±standard error (SE), and p<0.05 was considered significant.

### Body length, body mass, energy intake, torpor and activity time

We used a linear model to compare body mass and body length at time of separation from the mothers, that is, early post-weaning phase, between groups (EB vs. LB). Linear models were also used to assess differences between pre-hibernation body length of EB and LB juveniles and adult female dormice. Because responses of body mass, body length, activity time and all torpor parameters seemed to be composed of two piecewise linear segments, we examined the time course of each parameter by individual two-segment linear regressions. This method also allowed us to compare animal groups over the entire experimental period despite the fact that the length of the experimental periods differed markedly between EB and LB individuals. We determined the response variables by identifying the breaking point ('time of plateau' in case of body length and body mass, 'starting time' in case of activity time and torpor parameters; start of plateauing, increase or decrease of the respective parameter) visually and thereafter estimating the slope ('slope') of the linear part as well as the maximal level at which the parameter plateaued at the time of the break ('plateau'; only in case of body length and body mass) for each animal. Then, we performed linear mixed-effects models (lme, R package nlme, Pinheiro, Bates, DebRoy, Sarkar, and and the *R Development Core Team, 2014*) to analyze the effect of the interaction between group (EB vs. LB) and diet (AL vs. IF) on the time of plateau or starting time, the slope and the plateau (in case of body length and body mass) of each parameter. For activity time, the mean torpor duration was also added as explanatory variable. Beside the fixed effects, we included mean $T_a$ as random effect in all models to adjust for temperature-dependent variation. In addition, we analyzed the diet effect on torpor and activity times within each group by computing generalized linear mixed models (GLMMs) fitted by penalized quasi-likelihood procedure ('GLMM-PQL' in the R package MASS, *Ripley et al., 2014*). Other models were computed using the Poisson family with a log-link function for all torpor parameters and a Gaussian error distribution (identity-link) for energy intake and activity time. For all GLMMs, we investigated the main effects of diet and time (week as a factor) as well as their interaction as fixed factors and mean $T_a$ and animal-ID as random factors to adjust for temperature-dependent variation and repeated measurements within individuals. For activity time as response variable, the mean torpor duration and the energy intake were included as explanatory

variables. In models with torpor parameters as response variable, we entered energy intake as a fixed factor. Similarly, in models with energy intake as response variable, body mass was added as fixed factor. In addition, we analyzed the cumulative energy intake ('Ecum') at the end of the study periods (EB week 12 and LB week 8) as well as the mean cumulative energy consumption ('Ecum-mean' as Ecum of week 12 for EB and of week 8 for LB divided by the number of weeks, respectively) by computing linear models with group, diet, and the group-diet interaction as well as body mass as explanatory variables.

### Metabolic rate
We used linear mixed-effects models (*Pinheiro et al., 2014*) to analyze the effects of group, diet and time on metabolic rate variables, that is, ADMR, MR-day and MR-night. Body mass and animal-ID were added as random effects in these models to adjust for body-mass-dependent variation and repeated measurements within individuals. Owing to the limited sample size, and in order to avoid over-fitting, we restricted the statistical models to the main effects only, without including interaction terms.

### Body composition, total energy expenditure and water turnover
We used linear mixed-effects models to assess effects of time (pre vs. post-hibernation), group (EB vs. LB) and diet treatment (AL vs. IF) on body mass, FM and FFM levels of juvenile garden dormice, with individuals as a random factor. A linear model was also used to determine effects of group on percentage mass loss over hibernation. We also employed Tukey's post-hoc tests to reveal specific differences in BC between groups and diets. We tested the effects of group and diet on pre-hibernation TEE and $rH_2O$ by using linear model, also including body mass as an explanatory variable.

### Hibernating patterns
We used linear models to test for the effects of group and diet on hibernation duration, mean torpor and mean arousal durations, and minimal body temperature of juvenile garden dormice during winter. Further, significant effects of group and diet on arousal frequency were tested by using a generalized linear model with Poisson distribution. Hibernation duration was also added to the model as a covariate for arousal frequency.

### Body mass changes before breeding and reproductive success
We used a linear mixed-effects model to assess effects of time (post-hibernation vs. start of early breeding vs. start of late breeding), group (EB vs. LB) and diet treatment (AL vs. IF) on levels of body mass of juvenile dormice, with individual as a random factor. We also employed Tukey's post-hoc tests to reveal specific differences in body mass between times, groups and diets. Linear models also used to test for effects of groups and diet treatments on changes of body mass between post-hibernation and the start of early breeding, and between the start of early breeding and the start of late breeding. A generalized linear model with binomial distribution was used to test effects of the timing of breeding (early vs. late reproductive event), group and diet, with individuals as a random factor, on the breeding probability of female garden dormice during the subsequent reproductive year. We also used linear models to determine potential differences in litter size and litter mass between the female groups according to their time of birth and feeding treatment. Owing to the limited sample size, and in order to avoid over-fitting, we restricted the statistical models to the main effects only, without including interaction terms.

## Acknowledgements
The authors thank Peter Steiger for his help with animal care, Gerhard Fluch for building the loggers, and Sebastian Vetter for his advises in data analyses. The authors also want to acknowledge Walter Arnold, Franz Hölzl and Sebastian Vetter for inspiring and valuable discussions.

## Additional information

### Funding

| Funder | Grant reference number | Author |
|---|---|---|
| University of Veterinary Medicine Vienna | Postdoctoral Fellowship | Sylvain Giroud |
| Austrian Science Fund | P27267-B25 | Sylvain Giroud |

The funders had no role in study design, data collection and interpretation, or the decision to submit the work for publication.

### Author contributions

Britta Mahlert, Data curation, Formal analysis, Validation, Investigation, Visualization, Methodology, Writing—original draft, BM analysed the data, produced the figures and wrote the manuscript, BM carried out the laboratory work and processed the raw data; Hanno Gerritsmann, Resources, Investigation, HG performed surgical implantations of loggers and post-surgical animals care; Gabrielle Stalder, Resources, Investigation, GS performed surgical implantations of loggers and post-surgical animals care; Thomas Ruf, Software, Formal analysis, Writing—review and editing, TR helped with data processing and analyses; Alexandre Zahariev, Resources, Validation, Methodology, AZ performed stable isotopes analyses; Stéphane Blanc, Resources, Formal analysis, Validation, Methodology, Writing—review and editing, SB validated stable isotopes analyses; Sylvain Giroud, Conceptualization, Resources, Formal analysis, Supervision, Funding acquisition, Validation, Investigation, Methodology, Writing—original draft, Project administration, Writing—review and editing, SG conceived and designed the study, SG analysed the data, produced the figures and wrote the manuscript, SG carried out the laboratory work and processed the raw data

### Author ORCIDs

Britta Mahlert (ID) https://orcid.org/0000-0001-6598-9129
Sylvain Giroud (ID) https://orcid.org/0000-0001-6621-7462

### Ethics

Animal experimentation: All procedures were approved by the institutional ethics committee and the national authority according to §26 of Law for Animal Experiments, Tierversuchsgesetz 2013 - TGV 2013 (BMF - 68.205/0125-II/3b/2013)

### Decision letter and Author response

Decision letter https://doi.org/10.7554/eLife.31225.022
Author response https://doi.org/10.7554/eLife.31225.023

## Additional files

### Supplementary files

• Supplementary file 1. Parameters of linear models for the effects of group and diet on total energy expenditure and water turnover of juvenile garden dormice. Body mass was also assessed as an explanatory variable in the models. p-Values shown in bold correspond to statistically significant and interpretable values.
DOI: https://doi.org/10.7554/eLife.31225.012

• Supplementary file 2. Means ± standard errors and parameters of analyses of variance for the effects of group (early-born 'EB' vs. late-born 'LB'), diet (ad-libitum 'AL' vs. intermittently fasted 'IF') and time ('start of growth' vs. 'body mass plateau') on average daily metabolic rate ('ADMR') of garden dormice, metabolic rate during night ('MR-night') and metabolic rate during day ('MR-day'). Metabolic rate is expressed in ml $O_2$ $h^{-1}$ $g^{-1}$. Body mass and animal ID were included as random effects in all models. p-Values shown in bold correspond to statistically significant and interpretable values.

DOI: https://doi.org/10.7554/eLife.31225.013

• Supplementary file 3. Means and standard errors for the time ('starting time') and rate ('slope') of increase in torpor frequency, mean and total torpor duration of early-born ('EB') and late-born ('LB') juvenile garden dormice, either fed *ad libitum* ('AL') or intermittently fasted ('IF').

DOI: https://doi.org/10.7554/eLife.31225.014

• Supplementary file 4. Parameters of linear models for the effects of group and diet on the hibernation duration, arousal frequency, mean torpor and arousal durations and minimal body temperature ($T_b$) during winter. Hibernation duration was also assed as an explanatory variable in the model for arousal frequency. p-Values shown in italic correspond to statistically significant and interpretable values.

DOI: https://doi.org/10.7554/eLife.31225.015

• Supplementary file 5. Means and standard deviations for the hibernation duration, arousal frequency, mean torpor and arousal durations and minimal body temperature ($T_b$) of juveniles during winter, according to their time of birth (early-born 'EB', late-born 'LB') and feeding treatment (*ad libitum* 'AL' and intermittently fasted 'IF').

DOI: https://doi.org/10.7554/eLife.31225.016

• Supplementary file 6. Parameters of analyses of variance for the effects of time, group and diet on body mass and body mass changes of juvenile garden dormice during the post-hibernation breeding period. Body mass changes were computed between post-hibernation and the start of early breeding, and between the start of early breeding and the start of late breeding. p-Values shown in bold correspond to statistically significant and interpretable values.

DOI: https://doi.org/10.7554/eLife.31225.017

• Supplementary file 7. Means and standard deviations of body mass at post-hibernation, start of early breeding and start of late breeding, as well as body mass changes between pre-hibernation and the start of early breeding, and between the start of early breeding and the start of late breeding of the experimental animal groups, according to the time of birth (early-born 'EB', late-born 'LB') and feeding treatment (*ad libitum* 'AL' and intermittently fasted 'IF'). Groups differing significantly (p<0.05, Tukey's post-hoc comparisons) are denoted by different superscripts.

DOI: https://doi.org/10.7554/eLife.31225.018

• Supplementary file 8. Parameters of generalized linear models (in the case of breeding proportion) or linear models (in the case of litter size and litter mass) for the effects of timing of reproduction, group and diet on the female breeding proportion, litter size and litter mass. p-Values shown in bold correspond to statistically significant and interpretable values.

DOI: https://doi.org/10.7554/eLife.31225.019

• Transparent reporting form

DOI: https://doi.org/10.7554/eLife.31225.020

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
