## [Decision Letter]

Thank you for submitting your article "Implications of being born late in the active season for growth, fattening, torpor use, winter survival and fertility" for consideration by *eLife*. Your article has been reviewed by two peer reviewers, and the evaluation has been overseen by Diethard Tautz as the Senior and Reviewing Editor. The following individual involved in review of your submission has agreed to reveal his identity: Loren Buck (Reviewer #1).

The reviewers have discussed the reviews with one another and the Reviewing Editor has drafted this decision to help you prepare a revised submission.

The work provides a coherent overview of experiments on early and late born juveniles of great interest and novel insights. Detailed comments are provided in the attached reviews. The two major issues that need to be addressed are the function of pre-hibernation torpor in relation to fattening (reviewer 1) (specifically, is the expression of torpor functioning to preserve fat mass previously laid down or to facilitate further fattening?), and the point about post winter hibernation weight gain and success rate of conceiving (reviewer 2).

Note that the prior articles that referee 2 refers to are:

* Giroud et al. Late-born intermittently fasted juvenile garden dormice use torpor to grow and fatten prior to hibernation: consequences for ageing processes. Proc Biol Sci. 2014 Dec 22; 281(1797): 20141131. doi: 10.1098/rspb.2014.1131

* Stumpfel et al. Differences in growth rates and pre-hibernation body mass gain between early and late-born juvenile garden dormice. J Comp Physiol B. 2017; 187(1): 253-263. Published online 2016 Aug 1. doi: 10.1007/s00360-016-1017-x (Posted 5th Oct 2017)

Reviewer #1:

The submission entitled "Implications of being born late in the active season for growth, fattening, torpor use, winter survival and fertility" by Giroud et al. investigates the effects of timing of birth within a season (early or late born) and diet on growth, fattening, metabolic rate, patterns of torpor, survival and fecundity in captive female garden dormice. This is a very nice study builds logically on prior work by this group and is both novel and compelling in that it is the first investigation that I know of that specifically tests the impact of birth timing on expression of torpor. Moreover, findings related to the impacts of diet on fattening, metabolism, hibernation and reproduction in this species greatly increases our understanding of the interrelationships among these parameters shown in other hibernating species. The scientific approaches, methods and design are appropriate for the questions pursued. In addition, the statistical analyses appear rigorous and sound. The graphical and tabular representation of the data are clear. Specifically major findings that the authors report include that birth timing significantly impacted rate of growth (late born individuals grew twice as fast as early born), expression of torpor (late born individuals expressed torpor earlier and to a greater degree than early born) and that late born animals were compromised in their degree of fattening prior to entering hibernation. Interestingly, there were no differences between late born and early born animals in either their patterns of hibernation or timing of the end of their heterothermic season. Furthermore, late born females were more likely to reproduce in the following year.

A major conclusion the authors articulate in the Abstract is that torpor is incompatible with growth but promotes pre-hibernation fattening. This I find curious in that the data shows that 1) late born individuals express torpor earlier and to a greater degree than early born (Figure 4); 2) late born animals grow faster and to the same structural size as early born animals (Figure 1) and that 3) late born animals do not gain as much fat mass as their early born counterparts (Table 3). Unless I'm missing something here, it seems that early expression of torpor may more strongly facilitate growth while also facilitating (but to a lesser degree) fattening. However, the timing of the expression of torpor increasing in occurrence and duration happens after (or coincident with) animals have peaked in structural growth and body mass (hence fat mass). Quite likely, the expression of torpor functions to preserve fat mass previously laid down rather than to facilitate further fattening. This interpretation would be consistent with what Sheriff et al. report for arctic ground squirrels that reduce both body temperature and MR (both whole body and lean mass specific) in anticipation of hibernation. I strongly suggest the authors clearly address this criticism of the conclusions of this manuscript.

Reviewer #2:

The current study provides an extensive overview of physiological and life history parameters of juvenile female garden dormice exploring the effect of being born early vs. late in conjunction to diets with and without restricted food availability. The data confirm several previously reported observations (e.g. pre-hibernation BM, FM, torpor use and winter hibernation parameters) and substantiates novel findings on reproduction.

The strength of the paper, i.e. the integrated collection of various types of data throughout the animal's first year of life history, also embodies one of its weaknesses. It is difficult to extract a main message from the paper, particularly because many of the data have already been addressed in previous reports (albeit on different cohorts of animals). Nevertheless, the authors did a nice job in summarizing data in the conclusion section, in which also the novel findings are better highlighted than in the abstract. I would suggest the authors to focus on novel findings and discrepancies with previous studies – the 5 main figures basically represent duplications of previously published work, whereas the novel data are mainly presented in supplemental figures.

The absolutely novel finding constitutes of increased breeding success in LB versus EB dependent on diet. Rightfully, these data conflict with the silver-spoon concept. However, previous work shows that deleterious effects were strongest when the perceived time available for growth compensation prior to breeding was shortest. In this perspective, data on BM (energy intake, activity pattern, MR, etc.) in the post winter hibernation breeding animals would be of great interest. Diet of animals during post winter hibernation is unclear; were they still IF? Further, what do the bars on Figure 5 represent?

Secondly, there is a marked difference with previously published data on the source of body weight loss during winter hibernation. While previous studies in garden dormouse demonstrated a similar reduction in FMM and FM to account for BM loss, the current data almost exclusively point to FM loss. Can the authors offer any explanation for this?

Further, the relationship between various data may be clarified visually to the reader by combining multiple figures in panels of a single figure, e.g. Figure 1, Figure 2, Figure 4 and S2.

---

## [Author Response]

Reviewer #1:[…] A major conclusion the authors articulate in the Abstract is that torpor is incompatible with growth but promotes pre-hibernation fattening. This I find curious in that the data shows that 1) late born individuals express torpor earlier and to a greater degree than early born (Figure 4); 2) late born animals grow faster and to the same structural size as early born animals (Figure 1) and that 3) late born animals do not gain as much fat mass as their early born counterparts (Table 3). Unless I'm missing something here, it seems that early expression of torpor may more strongly facilitate growth while also facilitating (but to a lesser degree) fattening. However, the timing of the expression of torpor increasing in occurrence and duration happens after (or coincident with) animals have peaked in structural growth and body mass (hence fat mass). Quite likely, the expression of torpor functions to preserve fat mass previously laid down rather than to facilitate further fattening. This interpretation would be consistent with what Sheriff et al. report for arctic ground squirrels that reduce both body temperature and MR (both whole body and lean mass specific) in anticipation of hibernation. I strongly suggest the authors clearly address this criticism of the conclusions of this manuscript.

We agree with this view of the reviewer, in the sense that torpor is indeed greater after animals have peaked in structural growth and body mass (possibly acting to preserve fat mass previously laid down). However, we agree only partially because meanwhile torpor expression already started to increase in juveniles while they were still gaining mass during the last weeks, which mainly correspond to fattening, before their peak body mass. In contrast, torpor use in both early and late-born individuals stayed very low, if not zero, during the first weeks (to a body mass of approximately two thirds of the animals’ peak mass), which mainly corresponds to developmental growth. In particular, our data show that 1) late-born individuals express torpor earlier in time, but at a similar body mass (~80g) as early born individuals, and 2) late-born animals grow faster and to the same structural size as early-born, by means of higher energy intake, rather than by increased torpor use. According to a reviewer #2’s comment, this information on body length, body mass, total torpor use and energy intake was combined into a single figure, now Figure 1 in the manuscript. However, we agree with the major comment of reviewer #1 saying that torpor expression might also function to preserve fat content previously laid down. Hence we have now clearly addressed this criticism in the Abstract, Discussion and conclusion of the manuscript.

Reviewer #2:[…] The strength of the paper, i.e. the integrated collection of various types of data throughout the animal's first year of life history, also embodies one of its weaknesses. It is difficult to extract a main message from the paper, particularly because many of the data have already been addressed in previous reports (albeit on different cohorts of animals). Nevertheless, the authors did a nice job in summarizing data in the conclusion section, in which also the novel findings are better highlighted than in the abstract. I would suggest the authors to focus on novel findings and discrepancies with previous studies – the 5 main figures basically represent duplications of previously published work, whereas the novel data are mainly presented in supplemental figures.

We thank the reviewer for this comment. We agree that the absolute novel findings, i.e. increasing breeding success in LB vs. EB individuals, but also the better characterization of energetic responses of LB juveniles, were not emphasised enough in the manuscript. Therefore, the novel data, such as the proportion of breeders, energy intake, activity duration and overall energy expenditure, are now presented as main figures, and not anymore in supplementary material. Further, in several sections of the manuscript (for instance: subsection “Torpor use and activity time”, and subsection “Statistical analyses”), we have further discussed the novel findings and discrepancies with previous studies on garden dormice and in general on hibernators. The main findings are also better highlighted in the abstract.

The absolutely novel finding constitutes of increased breeding success in LB versus EB dependent on diet. Rightfully, these data conflict with the silver-spoon concept. However, previous work shows that deleterious effects were strongest when the perceived time available for growth compensation prior to breeding was shortest. In this perspective, data on BM (energy intake, activity pattern, MR, etc.) in the post winter hibernation breeding animals would be of great interest. Diet of animals during post winter hibernation is unclear; were they still IF? Further, what do the bars on Figure 5 represent?

We have analysed additional body mass data as suggested by the reviewer during the post-hibernation breeding period. We do not have any information for other variables, such as energy intake, activity patterns, MR etc. for the breeding periods. However, post-hibernation body mass analyses revealed no significant differences between either EB and LB individuals, or AL and IF animals. The data are now presented in Supplementary file 6 and Supplementary file 7 Paragraphs have been added in the sections ‘Statistical analyses’ (last paragraph) and ‘Results’ (subsection “Body mass during the subsequent reproductive period”) of the manuscript. During the post-winter hibernation period, all animals were provided with ad-libitum access to food and water. This information has been added to the subsection “Reproduction and fecundity”. Bars in Figure 5 represent standard deviations. This information has been added in the legend for Figure 5.

Secondly, there is a marked difference with previously published data on the source of body weight loss during winter hibernation. While previous studies in garden dormouse demonstrated a similar reduction in FMM and FM to account for BM loss, the current data almost exclusively point to FM loss. Can the authors offer any explanation for this?

In our previous study, the body composition of (late-born) juvenile dormice prior to winter was determined at the end of the period of highest summer mass gain (see Figure 2 and ‘Protocol overview’ on page 2 in Giroud et al., 2014), at a body mass (BM) of ~80g, which was ~10% lower than the peak BM (~90g) at entrance into hibernation. Since the last phase of BM gain prior to hibernation corresponds mainly to fattening, it is likely that pre-hibernation fat mass (FM) in juveniles was largely underestimated in our previous study. In contrast, the body composition of juvenile dormice in the present study was assessed slightly after the peak of pre-hibernation body mass. Hence the body composition, notably the FM level, is closer to the ‘optimal’ body composition of the animals at pre-hibernation. Indeed, in Giroud et al., 2014, ‘pre-hibernation’ FM and fat-free mass (FFM) corresponded to ~34% and ~66% of BM, respectively. In Mahlert et al., pre-hibernation FM and FFM represent ~42% and ~58% of BM, respectively, in late-born juveniles, and ~48% and 52% of BM, respectively, in early-born individuals. This methodological issue is likely to explain the marked difference between the data on the source of body mass loss in the present study compared with previously published data from Giroud et al., 2014. Further, in the present study, body composition at both the onset and emergence from hibernation was determined by using the same method of isotopic dilution. It was not the same in Giroud et al., 2014, in which body composition was determined by means of two different methods, i.e. isotopic dilution prior to winter and Soxhlet method after hibernation. For all these reasons, we are now quite confident with our current data (that almost exclusively point to FM loss overwinter) to have rigorously assessed the body composition of the animals at onset and emergence from hibernation, hence to have correctly determined the source (i.e. FM and FFM) of body mass loss during winter hibernation. We now offer an explanation for this discrepancy with our previous published data in the last paragraph of the subsection “Consequences for winter hibernation and substrate utilization over hibernation in juveniles”.

Further, the relationship between various data may be clarified visually to the reader by combining multiple figures in panels of a single figure, e.g. Figure 1, Figure 2, Figure 4 and S2.

We thank the reviewer for this suggestion. Figure 1, Figure 2, Figure 4 have been pooled together in a single figure, which now refers as Figure 1 of the manuscript. As a consequence, Figure S3 (on breeding occurrence) is now a main figure in the manuscript and refers now to Figure 5.